# Find Your Friends: Personalized Federated Learning with the Right Collaborators

## Abstract

In the traditional federated learning setting, a central server coordinates a network of clients to train one global model. However, the global model may serve many clients poorly due to data heterogeneity. Moreover, there may not exist a trusted central party that can coordinate the clients to ensure that each of them can benefit from others. To address these concerns, we present a novel decentralized framework, *FedeRiCo*, where each client can learn as much or as little from other clients as is optimal for its local data distribution. Based on expectation-maximization, FedeRiCo estimates the utilities of other participants' models on each client's data so that everyone can select the right collaborators for learning. As a result, our algorithm outperforms other federated, personalized, and/or decentralized approaches on several benchmark datasets, being the *only* approach that consistently performs better than training with local data only.

## 1 Introduction

Federated learning (FL) (McMahan et al., 2017) offers a framework in which a single server-side model is collaboratively trained across decentralized datasets held by clients. It has been successfully deployed in practice for developing machine learning models without direct access to user data, which is essential in highly regulated industries such as banking and healthcare (Long et al., 2020; Sadilek et al., 2021). For example, several hospitals that each collect patient data may want to merge their datasets for increased diversity and dataset size but are prohibited due to privacy regulations.

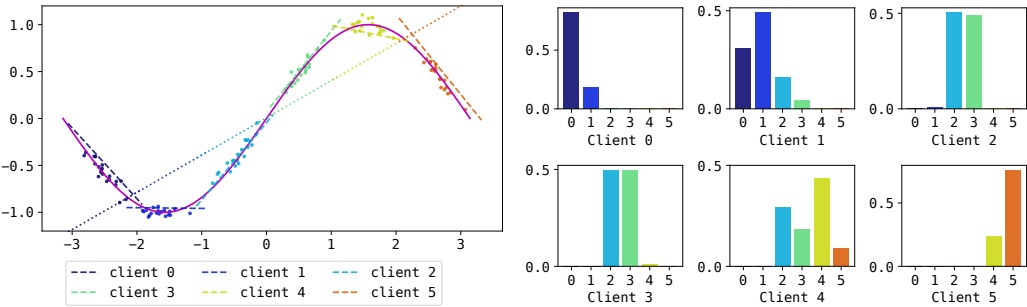

Figure 1: **Left:** Noisy data points generated for each client along a sine curve (solid magenta line) where the $x$-axis and $y$-axis correspond to input and output respectively. The corresponding model learned by FedAvg (dotted line) fails to adapt to the local data seen by each client, in contrast to the models learned by each client using our FedeRiCo (dashed lines). **Right:** The weights used by FedeRiCo to average participant outputs for each client. As the client index increases, the data is generated from successive intervals of the sine curve, and collaborator weights change accordingly.

Traditional FL methods like Federated Averaging (FedAvg) (McMahan et al., 2017) can achieve noticeable improvement over local training when the participating clients' data are homogeneous. However, each client's data is likely to have a different distribution from others in practice (Zhao et al., 2018; Adnan et al., 2022). Such differences make it much more challenging to learn a global model that works well for all participants. As an illustrative example, consider a simple scenario

where each client seeks to fit a linear model to limited data, on an interval of the sine curve as shown in Fig. 1. This is analogous to the FL setting where several participating clients would like to collaborate, but each client only has access to data from its own data distribution. It is clear that no single linear model can be adequate to describe the entire joint dataset, so a global model learned by FedAvg can perform poorly, as shown by the dotted line. Ideally, each client should benefit from collaboration by increasing the effective size and diversity of data, but in practice, forcing everyone to use the same global model without proper personalization can hurt performance on their own data distribution (Kulkarni et al., 2020; Tan et al., 2022).

To address this, we propose **Fede**rating with the **Ri**ght **Co**llaborators (FedeRiCo), a novel framework suitable for every client to find other participants with similar data distributions to collaborate with. Back to our illustration in Fig. 1. FedeRiCo enables each client to choose the *right collaborators* as shown on the plots on the right-hand side: each client is able to correctly leverage information from the neighboring clients when it is beneficial to do so. The final personalized models can serve the local distributions well, as demonstrated in the left plot.

More specifically, our FedeRiCo assumes that each client has an underlying data distribution, and exploits the hidden relationship among the clients' data. By selecting the most relevant clients, each client can collaborate as much or as little as they need, and learn a personalized mixture model to fit the local data. Additionally, FedeRiCo achieves this in a fully decentralized manner that is not beholden to any central authority (Li et al., 2021a; Huang et al., 2021; Kalra et al., 2021).

**Our contributions**   We propose FedeRiCo, a novel decentralized and personalized FL framework derived based on expectation-maximization (EM). Within this framework, we propose a communication-efficient protocol suitable for fully-decentralized learning. Through extensive experiments on several benchmark datasets, we demonstrate that our approach finds good client collaboration and outperforms other methods in the non-i.i.d. data distributions setting.

**Paper outline**   The rest of the paper is organized as follows. In Section 2 we discuss related approaches towards decentralized federated learning and personalization. Section 3 describes our algorithm formulation and its relationship to expectation-maximization, and an efficient protocol for updating clients. We provide experimental results in Section 4, and conclude in Section 5.

## 2   RELATED WORK FOR PERSONALIZED FL

**Meta-learning**   Federated learning can be interpreted as a meta-learning problem, where the goal is to extract a global meta-model based on data from several clients. This meta-model can be learned using, for instance, the well-known Federated Averaging (FedAvg) algorithm (McMahan et al., 2017), and personalization can then be achieved by locally fine-tuning the meta-model (Jiang et al., 2019). Later studies explored methods to learn improved meta-models. Khodak et al. (2019) proposed ARUBA, a meta-learning algorithm based on online convex optimization, and demonstrates that it can improve upon FedAvg's performance. Per-FedAvg (Fallah et al., 2020) uses the Model Agnostic Meta-Learning (MAML) framework to build the initial meta-model. However, MAML requires computing or approximating the Hessian term and can therefore be computationally prohibitive. Acar et al. (2021) adopted gradient correction methods to explicitly de-bias the meta-model from the statistical heterogeneity of client data and achieved sample-efficient customization of the meta-model.

**Model regularization / interpolation**   Several works improve personalization performance by regularizing the divergence between the global and local models (Hanzely & Richtárik, 2020; Li et al., 2021b; Huang et al., 2021). Similarly, PFedMe (T Dinh et al., 2020) formulates personalization as a proximal regularization problem using Moreau envelopes. FML (Shen et al., 2020) adopts knowledge distillation to regularize the predictions between local and global models and handle model heterogeneity. In recent work, SFL (Chen et al., 2022) also formulates the personalization as a bi-level optimization problem with an additional regularization term on the distance between local models and its neighbor models according to a connection graph. Specifically, SFL adopts GCN to represent the connection graph and learns the graph as part of the optimization to encourage useful client collaborations. Introduced by Mansour et al. (2020) as one of the three methods for achieving personalization in FL, model interpolation involves mixing a client's local model with a jointly

trained global model to build personalized models for each client. Deng et al. (2020) further derive generalization bounds for mixtures of local and global models.

**Multi-task learning** Personalized FL naturally fits into the multi-task learning (MTL) framework. MOCHA (Smith et al., 2017) utilizes MTL to address both systematic and statistical heterogeneity but is restricted to simple convex models. VIRTUAL (Corinzia et al., 2019) is a federated MTL framework for non-convex models based on a hierarchical Bayesian network formed by the central server and the clients, and inference is performed using variational methods. SPO (Cui et al., 2021) applies Specific Pareto Optimization to identify the optimal collaborator sets and learn a hypernetwork for all clients. While also aiming to identify necessary collaborators, SPO adopts a centralized FL setting with clients jointly training the hypernetwork. In contrast, our work focuses on decentralized FL where clients aggregate updates from collaborators, and jointly make predictions.

In a similar spirit to our work, Marfoq et al. (2021) assume that the data distribution of each client is a mixture of several underlying distributions/components. Federated MTL is then formulated as a problem of modeling the underlying distributions using Federated Expectation-Maximization (FedEM). Clients jointly update a set of several component models, and each maintains a customized set of weights, corresponding to the mixing coefficients of the underlying distributions, for predictions. One shortcoming of FedEM is that it uses an instance-level weight assignment in training time but a client-level weight assignment in inference time. As a concrete example, consider a client consisting of a 20%/80% data mixture from distributions A and B. FedEM will learn two models, one for each distribution. Given a new data point at inference time, the client will always predict $0.2 \cdot \text{pred}_A + 0.8 \cdot \text{pred}_B$, *regardless of whether it came from distribution A or B*. This is caused by the mismatched behaviour between training and inference time. On the contrary, FedeRiCo naturally considers a client-level weight assignment for both training and inference in a decentralized setting.

**Other approaches** Clustering-based approaches are also popular for personalized FL (Sattler et al., 2020; Ghosh et al., 2020; Mansour et al., 2020). Such personalization lacks flexibility since each client can only collaborate with other clients within the same cluster. FedFomo (Zhang et al., 2021) interpolates the model updates of each client with those of other clients to improve local performance. FedPer (Arivazhagan et al., 2019) divides the neural network model into base and personalization layers. Base layers are trained jointly, whereas personalization layers are trained locally.

## 3 FEDERATED LEARNING WITH THE RIGHT COLLABORATORS

### 3.1 PROBLEM FORMULATION

We consider a federated learning (FL) scenario with $K$ clients. Let $[K] := \{1, 2, \ldots, K\}$ denote the set of positive integers up until $K$. Each client $i \in [K]$ consists of a local dataset $D_i = \{(\mathbf{x}_s^{(i)}, y_s^{(i)})\}_{s=1}^{n_i}$ where $n_i$ is the number of examples for client $i$, and the input $\mathbf{x}_s \in \mathcal{X}$ and output $y_s \in \mathcal{Y}$ are drawn from a joint distribution $\mathcal{D}_i$ over the space $\mathcal{X} \times \mathcal{Y}$.

The goal of personalized FL is to find a prediction model $h_i : \mathcal{X} \mapsto \mathcal{Y}$ that can perform well on the local distribution $\mathcal{D}_i$ for each client. One of the main challenges in personalized FL is that we do not know if two clients $i$ and $j$ share the same underlying data distribution. If their data distributions are vastly different, forcing them to collaborate is likely to result in worse performance compared to local training without collaboration. Our method, **Fede**rating with the **Ri**ght **Co**llaborators (FedeRiCo), is designed to address this problem so that each client can choose to collaborate or not, depending on their data distributions. FedeRiCo is a decentralized framework (i.e. without a central server). For better exposition, Section 3.2 first demonstrates how our algorithm works in a *hypothetical* all-to-all communication setting, an assumption that is then removed in Section 3.3 which presents several practical considerations for FedeRiCo to work with limited communication.

### 3.2 FEDERICO WITH ALL-TO-ALL COMMUNICATION

Note that every local distribution $\mathcal{D}_i$ can always be represented as a mixture of $\{\mathcal{D}_j\}_{j=1}^K$ with some client weights $\boldsymbol{\pi}_i = [\pi_{i1}, \ldots, \pi_{iK}] \in \Delta^K$, where $\Delta^K$ is the $(K-1)$-dimensional

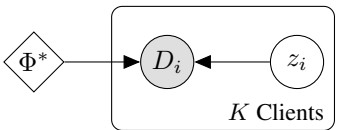

Figure 2: Graphical model

simplex[1]. Let $z_i$ be the latent assignment variable of client $i$, and $\Pi := [\boldsymbol{\pi}_1, \ldots, \boldsymbol{\pi}_K]^\top$ be the prior $\Pi_{ij} = \Pr(z_i = j)$. Suppose that the conditional probability $p_i(y|\mathbf{x})$ satisfies $-\log p_i(y|\mathbf{x}) = \ell(h_{\boldsymbol{\phi}_i^*}(\mathbf{x}), y) + c$ for some parameters $\boldsymbol{\phi}_i^* \in \mathbb{R}^d$, loss function $\ell : \mathcal{Y} \times \mathcal{Y} \mapsto \mathbb{R}^+$, and normalization constant $c$. By using the stacked notation $\Phi^* = [\boldsymbol{\phi}_1^*, \ldots, \boldsymbol{\phi}_K^*] \in \mathbb{R}^{d \times K}$, Fig. 2 shows the graphical model of how the local dataset is generated. Our goal is to learn the parameters $\Theta := (\Phi, \Pi)$ by maximizing the log-likelihood:

$$f(\Theta) := \frac{1}{n} \log p(D; \Theta) = \frac{1}{n} \sum_{i=1}^{K} \log p(D_i; \Theta) = \frac{1}{n} \sum_{i=1}^{K} \log \sum_{z_i=1}^{K} p(D_i, z_i; \Theta). \tag{1}$$

where $D := \cup_i D_i$ and $n := \sum_i n_i$. One standard approach to optimization with latent variables is expectation maximization (EM) (Dempster et al., 1977). The corresponding variational lower bound is given by (all detailed derivations of this section can be found in Appendix A)

$$\mathcal{L}(q, \Theta) := \frac{1}{n} \sum_i \mathbb{E}_{q(z_i)}[\log p(D_i, z_i; \Theta)] + C, \tag{2}$$

where $C$ is a constant not depending on $\Theta$. To obtain concrete objective functions suitable for optimization, we further assume that $p_i(x) = p(x), \forall i \in [K]$. Similar to Marfoq et al. (2021), this assumption is required due to technical reasons and can be relaxed if needed. With this assumption, we perform the following updates at each iteration $t$:

- **E-step:** For each client, find the best $q$, which is the posterior $p(z_i = j|D_i; \Theta^{(t-1)})$ given the current parameters $\Theta^{(t-1)}$:

$$w_{ij}^{(t)} := q^{(t)}(z_i = j) \propto \Pi_{ij}^{(t-1)} \exp\left[ -\sum_{s=1}^{n_i} \ell\left( h_{\boldsymbol{\phi}_j^{(t-1)}}(\mathbf{x}_s^{(i)}), y_s^{(i)} \right) \right]. \tag{3}$$

- **M-step:** Given the posterior $q^{(t)}$ from the E-step, maximize $\mathcal{L}$ w.r.t. $\Theta = (\Phi, \Pi)$:

$$\Pi_{ij}^{(t)} = w_{ij}^{(t)} \qquad \text{and} \qquad \Phi^{(t)} \in \operatorname*{argmin}_{\Phi} \frac{1}{n} \sum_{i=1}^{K} \widehat{\mathcal{L}}_{w,i}(\Phi) \tag{4}$$

$$\text{where} \qquad \widehat{\mathcal{L}}_{w,i}(\Phi) := \sum_{j=1}^{K} w_{ij}^{(t)} \sum_{s=1}^{n_i} \ell\left( h_{\boldsymbol{\phi}_j}(\mathbf{x}_s^{(i)}), y_s^{(i)} \right). \tag{5}$$

Bear in mind that each client can only see its local data $D_i$ in the federated setting. The E-step is easy to compute once the models from other clients $\boldsymbol{\phi}_j, j \neq i$ are available. $\Pi_{ij}^{(t)}$ is also easy to obtain as the posterior $w_{ij}^{(t)}$ is stored locally. However, $\Phi^{(t)}$ is trickier to compute since each client can potentially update $\Phi$ towards different directions due to data heterogeneity amongst the clients. To stabilize optimization and avoid overfitting from client updates, we rely on small gradient steps in lieu of full optimization in each round. To compute $\Phi^{(t)}$ algorithmically, each client $i$:

1. Fixes $w_{ij}^{(t)}$ and computes the local gradient $\nabla\widehat{\mathcal{L}}_{w,i}(\Phi^{(t-1)})$ on local $D_i$.
2. Broadcasts $\nabla\widehat{\mathcal{L}}_{w,i}(\Phi^{(t-1)})$ to and receives $\nabla\widehat{\mathcal{L}}_{w,j}(\Phi^{(t-1)})$ from other clients $j \neq i$. The models are updated based on the aggregated gradient with step size $\eta > 0$:

$$\Phi^{(t)} = \Phi^{(t-1)} - \eta \sum_{j=1}^{K} \nabla\widehat{\mathcal{L}}_{w,j}(\Phi^{(t-1)}). \tag{6}$$

Each client uses $\widehat{h}_i(\mathbf{x}) = \sum_j w_{ij}^{(t)} h_{\boldsymbol{\phi}_j^{(t)}}(\mathbf{x})$ for prediction after convergence.

**Remark 1** The posterior $w_{ij}^{(t)}$ (or equivalently the prior in the next iteration $\Pi_{ij}^{(t)}$) reflects the importance of model $\boldsymbol{\phi}_j$ on the data $D_i$. When $w_{ij}^{(t)}$ is one-hot with a one in the $i$th position, client $i$ can perform learning by itself without collaborating with others. When $w_{ij}^{(t)}$ is more diverse, client $i$

---

[1]One-hot $\boldsymbol{\pi}_i$ is always feasible, but other mixing coefficients may exist.

can find the right collaborators with useful models $\phi_j$. Such flexibility enables each client to make its own decision on whether or not to collaborate with others, hence the name of our algorithm.

**Remark 2** Unlike prior work (Mansour et al., 2020; Marfoq et al., 2021), our assignment variable $z$ and probability $\Pi$ are on the client level. If we assume that all clients share the same prior (i.e., there is only a vector $\boldsymbol{\pi}$ instead of a matrix $\Pi$), the algorithm would be similar to HypCluster (Mansour et al., 2020). Marfoq et al. (2021) used a similar formulation as ours but their assignment variable $z$ is on the instance level: every data point (instead of client) comes from a mixture of distributions. Such an approach can cause several issues at inference time, as the assignment for novel data point is unknown. We refer the interested readers to Section 2 and Section 4 for further comparison.

**Theoretical Convergence** Under some regularity assumptions, our algorithm converges as follows:

**Theorem 3.1.** *[Convergence] Under Assumptions E.1-E.6, when the clients use SGD with learning rate $\eta = \frac{a_0}{\sqrt{T}}$, and after sufficient rounds $T$, the iterates of our algorithm satisfy*

$$\frac{1}{T}\sum_{t=1}^{T}\mathbb{E}\|\nabla_\Phi f(\Phi^t, \Pi^t)\|_F^2 \leq \mathcal{O}\left(\frac{1}{\sqrt{T}}\right), \qquad \frac{1}{T}\sum_{t=1}^{T}\Delta_\Pi f(\Phi^t, \Pi^t) \leq \mathcal{O}\left(\frac{1}{T^{3/4}}\right), \qquad (7)$$

*where the expectation is over the random batch samples and $\Delta_\Pi f(\Phi^t, \Pi^t) := f(\Phi^t, \Pi^t) - f(\Phi^t, \Pi^{t+1}) \geq 0$.*

Due to space limitations, further details and the complete proof are deferred to Appendix E. The above theorem shows that the gradient w.r.t. the model parameters $\Phi$ and the improvement over the mixing coefficients $\Pi$ becomes small as we increase the round $T$, thus converging to a stationary point of the log-likelihood objective $f$.

## 3.3 COMMUNICATION-EFFICIENT PROTOCOL

So far, we have discussed how FedeRiCo works in the all-to-all communication setting. In practice, FedeRiCo does not require excessive model transmission, and this subsection discusses several practical considerations to ensure communication efficiency. Specifically, we tackle the bottlenecks in both the E-step (3) and the M-step (4) since they require joint information of all models $\Phi$. Due to space constraint, the pseudocode is provided in Algorithm 1, Appendix B.

**E-step** For client $i$, the key missing quantity to compute (3) without all-to-all communication is the loss $\ell(\phi_j^{(t-1)})$, or likelihood $p(D_i|z_i = j; \Phi^{(t-1)})$, of other clients' models $\phi_j, j \neq i$. Since the models $\Phi$ are being updated slowly, one can expect that $\ell(\phi_j^{(t-1)})$ will not be significantly different from the loss $\ell(\phi_j^{(t-2)})$ of the previous iteration. Therefore, each client can maintain a list of losses for all the clients, sample a subset of clients in each round using a sampling scheme $\mathcal{S}$ (e.g., $\epsilon$-greedy sampling as discussed later), and only update the losses of the chosen clients.

**M-step** To clearly see how $\Phi$ is updated in the M-step, let's focus on the update to a specific client's model $\phi_i$. According to (5) and (6), the update to $\phi_i$ is given by

$$-\eta \sum_{j=1}^{K} w_{ji}^{(t)} \sum_{s=1}^{n_j} \nabla_{\phi_i}\ell\left(h_{\phi_i}(\mathbf{x}_s^{(j)}), y_s^{(j)}\right). \qquad (8)$$

Note that the aggregation is based on $w_{ji}^{(t)}$ instead of $w_{ij}^{(t)}$. Intuitively, this suggests $\phi_i$ should be updated based on how the model is being used by *other clients* rather than how client $i$ itself uses it. If $\phi_i$ does not appear to be useful to all clients, i.e. $w_{ji}^{(t)} = 0, \forall j$, it does not get updated. Therefore, whenever client $i$ is sampled by another client $j$ using the sampling scheme $\mathcal{S}$, it will send $\phi_i$ to $j$, and receives the gradient update $\mathbf{g}_{ij} := w_{ji}^{(t)}\sum_{s=1}^{n_j}\nabla_{\phi_i}\ell\left(h_{\phi_i}(\mathbf{x}_s^{(j)}), y_s^{(j)}\right)$ from client $j$. One issue here is that $\mathbf{g}_{ij}$ is governed by $w_{ji}^{(t)}$, which could be arbitrarily small, leading to no effective update to $\phi_i$. We will show how this can be addressed by using an $\epsilon$-greedy sampling scheme.

**Sampling scheme $\mathcal{S}$** We deploy an $\epsilon$-greedy scheme where, in each round, each client uniformly samples clients with probability $\epsilon \in [0, 1]$ and samples the client(s) with the highest posterior(s) otherwise. This allows a trade off between emphasizing gradient updates from high-performing clients

(small $\epsilon$), versus receiving updates from clients uniformly to find potential collaborators (large $\epsilon$). The number $M$ of sampled clients (neighbors) per round and $\epsilon$ can be tuned based on the specific problem instance. We will show the effect of varying the hyperparameters in the experiments.

**Tracking the losses for the posterior** The final practical consideration is the computation of the posterior $w_{ij}^{(t)}$. From the E-step (3) and the M-step (4), one can see that $w_{ij}^{(t)}$ is the softmax transformation of the negative accumulative loss $L_{ij}^{(t)} := \sum_{\tau=1}^{t-1} \ell_{ij}^{(\tau)}$ over rounds (see Appendix A for derivation). However, the accumulative loss can be sensitive to noise and initialization. If one of the models, say $\phi_j$, performs slightly better than other models for client $i$ at the beginning of training, then client $i$ is likely to sample $\phi_j$ more frequently, thus enforcing the use of $\phi_j$ even when other better models exist. To address this, we instead keep track of the exponential moving average of the loss with a momentum parameter $\beta \in [0, 1)$, $\widehat{L}_{ij}^{(t)} = (1 - \beta)\widehat{L}_{ij}^{(t-1)} + \beta l_{ij}^{(t)}$, and compute $w_{ij}^{(t)}$ using $\widehat{L}_{ij}^{(t)}$. This encourages clients to seek new collaborators rather than focusing on existing ones.

## 4 EXPERIMENTS

### 4.1 EXPERIMENTAL SETTINGS

We conduct a range of experiments to evaluate the performance of our proposed FedeRiCo with multiple datasets. Additional experiment details and results can be found in Appendix D.

**Datasets** We compare different methods on several real-world datasets. We evaluate on image-classification tasks with the CIFAR-10, CIFAR-100 (Krizhevsky et al., 2009), and Office-Home[2] (Venkateswara et al., 2017) datasets. Particularly, we consider a non-IID data partition among clients by first splitting data by labels into several groups with disjoint label sets. Each group is considered a distribution, and each client samples from one distribution to form its local data. For each client, we randomly divide the local data into 80% training data and 20% test data.

**Baseline methods** We compare our FedeRiCo to several federated learning baselines. FedAvg (McMahan et al., 2017) trains a single global model for every client. We also compare to other personalized FL approaches including FedAvg with local tuning (FedAvg+) (Jiang et al., 2019), Clustered FL (Sattler et al., 2020), FedEM (Marfoq et al., 2021)[3], FedFomo (Zhang et al., 2021), as well as a local training baseline. All accuracy results are reported in mean and standard deviation across different random data split and random training seeds. Unless specified otherwise, we use 3 neighbors with $\epsilon = 0.3$ and momentum $\beta = 0.6$ as the default hyperparamters for FedeRiCo in all experiments. For FedEM, we use 4 components, which provides sufficient capacity to accommodate different numbers of label groups (or data distributions). For FedFomo, we hold out 20% of the training data for client weight calculations. For FedAvg+, we follow Marfoq et al. (2021) and update the local model with 1 epoch of local training.

**Training settings** For all models, we use the Adam optimizer with learning rate 0.01. CIFAR experiments use 150 rounds of training, while Office-Home experiments use 400 rounds. CIFAR-10 results are reported across 5 different data splits and 3 different training seeds for each data split. CIFAR-100 and Office-Home results are reported across 3 different data splits with a different training seed for each split.

### 4.2 PERFORMANCE COMPARISON

The performance of each FL method is shown in Table 1. Following the settings introduced by Marfoq et al. (2021), each client is evaluated on its own local testing data and the average accuracies weighted by local dataset sizes are reported. We observe that FedeRiCo has the best performance across all datasets and number of data distributions. Here, local training can be seen as an indicator to assess if other methods benefit from client collaboration as local training has no collaboration at all. We observe that our proposed FedeRiCo is the only method that consistently outperforms local training, meaning that FedeRiCo is the only method that consistently encourages effective

---

[2]This dataset has been made publically available for research purposes only.

[3]We use implementations from `https://github.com/omarfoq/FedEM` for Clustered FL and FedEM

Table 1: Accuracy (in percentage) with different number of data distributions. Best results in bold.

| | CIFAR-10 # of distributions | | | CIFAR-100 # of distributions | | | Office-Home # of distributions | | |
| Method | 2 | 3 | 4 | 2 | 3 | 4 | 2 | 3 | 4 |
| --- | --- | --- | --- | --- | --- | --- | --- | --- | --- |
| FedAvg | $11.44_{\pm 3.28}$ | $11.73_{\pm 3.68}$ | $13.93_{\pm 5.74}$ | $21.28_{\pm 5.04}$ | $17.41_{\pm 3.27}$ | $18.36_{\pm 3.68}$ | $66.58_{\pm 1.88}$ | $53.36_{\pm 4.21}$ | $51.25_{\pm 4.37}$ |
| FedAvg+ | $12.45_{\pm 8.46}$ | $29.86_{\pm 17.85}$ | $45.65_{\pm 21.61}$ | $29.95_{\pm 1.07}$ | $35.33_{\pm 1.77}$ | $36.17_{\pm 3.27}$ | $80.21_{\pm 0.68}$ | $81.88_{\pm 0.91}$ | $84.50_{\pm 1.37}$ |
| Local Training | $40.09_{\pm 2.84}$ | $55.27_{\pm 3.11}$ | $69.03_{\pm 7.05}$ | $16.60_{\pm 0.64}$ | $25.99_{\pm 2.38}$ | $31.05_{\pm 1.68}$ | $76.76_{\pm 0.23}$ | $83.30_{\pm 0.32}$ | $88.05_{\pm 0.44}$ |
| Clustered FL | $11.50_{\pm 3.65}$ | $15.24_{\pm 5.79}$ | $16.43_{\pm 5.17}$ | $20.93_{\pm 3.57}$ | $23.15_{\pm 7.04}$ | $15.15_{\pm 0.60}$ | $66.58_{\pm 1.88}$ | $53.36_{\pm 4.21}$ | $51.25_{\pm 4.37}$ |
| FedEM | $41.21_{\pm 10.83}$ | $55.08_{\pm 6.71}$ | $63.61_{\pm 9.93}$ | $26.25_{\pm 2.40}$ | $24.11_{\pm 7.36}$ | $19.23_{\pm 2.58}$ | $22.59_{\pm 1.95}$ | $28.72_{\pm 1.83}$ | $22.46_{\pm 3.99}$ |
| FedFomo | $42.24_{\pm 8.32}$ | $59.45_{\pm 5.57}$ | $71.05_{\pm 6.09}$ | $12.15_{\pm 0.57}$ | $20.49_{\pm 2.90}$ | $24.53_{\pm 2.77}$ | $78.61_{\pm 0.78}$ | $82.57_{\pm 0.24}$ | $87.86_{\pm 0.77}$ |
| FedeRiCo | $\mathbf{56.61}_{\pm 2.51}$ | $\mathbf{69.76}_{\pm 2.25}$ | $\mathbf{78.22}_{\pm 4.80}$ | $\mathbf{30.95}_{\pm 1.62}$ | $\mathbf{39.19}_{\pm 1.64}$ | $\mathbf{41.41}_{\pm 1.07}$ | $\mathbf{83.56}_{\pm 0.49}$ | $\mathbf{90.28}_{\pm 0.75}$ | $\mathbf{93.76}_{\pm 0.12}$ |

client collaborations. Notably, both FedEM and FedFomo performs comparably well to FedeRiCo on CIFAR-10 but worse when the dataset becomes more complex like CIFAR-100. This indicates that building the right collaborations among clients becomes a harder problem for more complex datasets. Moreover, FedEM can become worse as the number of distributions increases, even worse than local training, showing that it is increasingly hard for clients to participate effectively under the FedEM framework for complex problems with more data distributions.

In addition, Clustered FL has similar performance to FedAvg, indicating that it is hard for Clustered FL to split into the right clusters. In Clustered FL (Sattler et al., 2020), every client starts in the same cluster and cluster split only happens when the FL objective is close to a stationary point, i.e. the norm of averaged gradient update from all clients inside the cluster is small. Therefore, in a non-i.i.d setting like ours, the averaged gradient update might always be noisy and large, as clients with different distributions are pushing diverse updates to the clustered model. As a result, the cluster splitting rarely happens which makes clustered FL more like FedAvg.

## 4.3 CLIENT COLLABORATION

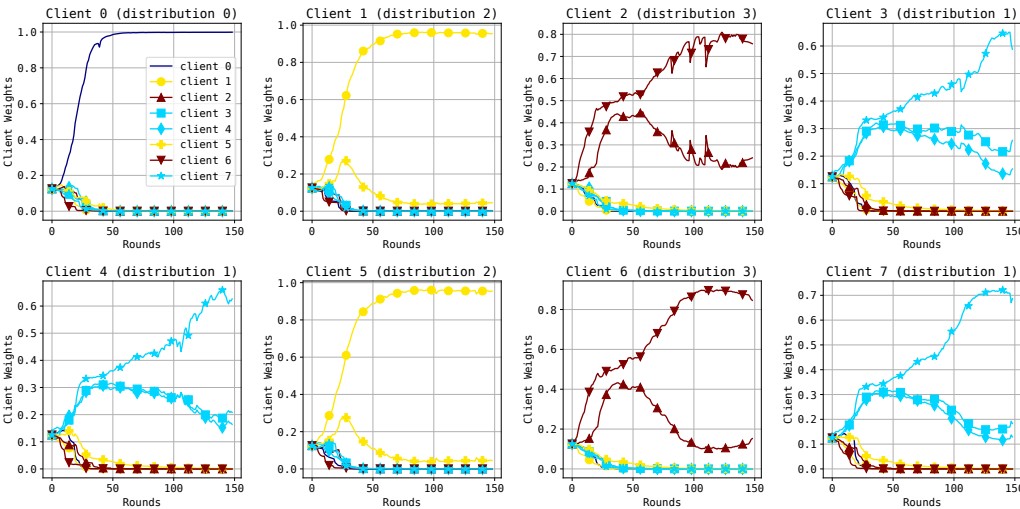

Figure 3: Client weights over time of FedeRiCo with CIFAR100 data and four different client distributions. Clients are color coded by their private data's distribution.

In this section, we investigate client collaboration by plotting the personalized client weights $w_{ij}^{(t)}$ of FedeRiCo over training. With different client data distributions, we show that FedeRiCo can assign more weight to clients from the same distribution. As shown in Fig. 3, we observe that clients with similar distributions collaborate to make the final predictions. For example, clients 3, 4 and 7 use a mixture of predictions from each other (in light blue) whereas client 0 only uses itself for prediction since it is the only client coming from distribution 0 (in dark blue) in this particular random split.

On the contrary, as shown in Fig. 4, even with 4 components, FedEM fails to use all of them for predictions for the 4 different data distributions. In fact, clients 2, 3, 4, 6 and 7 coming from two different distributions are using only the model of component 3 for prediction, whereas component 0 is never used by any client. Based on this, we find FedeRiCo better encourages the clients to

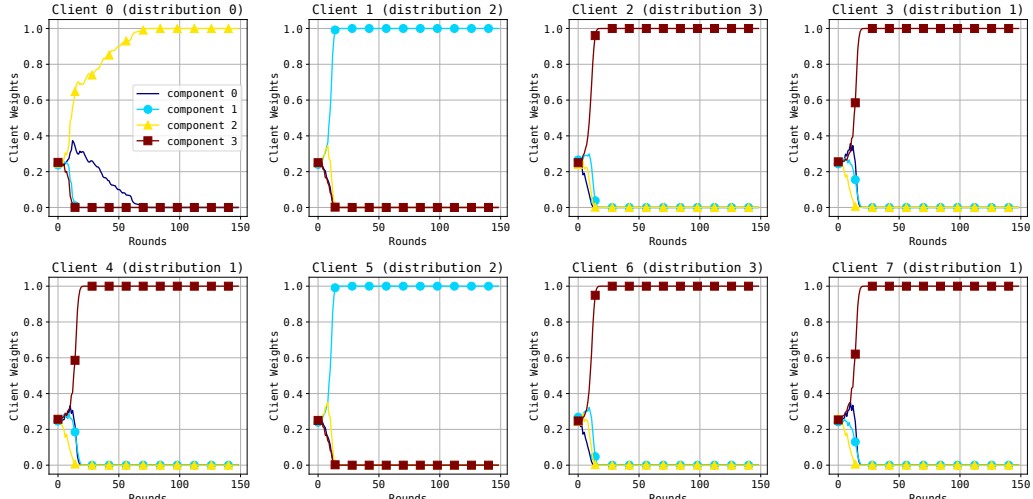

Figure 4: Component weights over training for FedEM with 4 components, on CIFAR100 data with 4 different client distributions. Clients are color coded by their private data's distribution.

collaborate with other similar clients and less with different clients. Each client can collaborate as much or as little as they need. Additionally, since all the non-similar clients have a weight of (almost) 0, each client only needs a few models from their collaborators for prediction.

## 4.4 EFFECT OF USING EXPONENTIAL MOVING AVERAGE LOSS

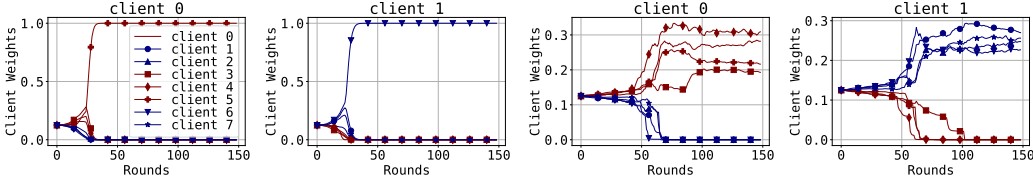

(a) Client weights with accumulative loss.      (b) Client weights with exponential moving average.

Figure 5: Effect on client weights with different implementations. The client weights on CIFAR-10 with 2 different client distributions are reported.

Here, we visualize the effect of using the exponential moving average loss by plotting client weights with both accumulative loss and exponential moving average loss in Fig. 5[4]. We observe that with the accumulative loss in Fig. 5a, the client weights quickly converge to one-hot, while with the exponential moving average loss in Fig. 5b, the client weights are more distributed to similar clients. This corresponds to our expectation stated in Section 3.3: the clients using exponential moving average loss are expected to seek for more collaboration compared to using accumulative loss.

## 4.5 HYPERPARAMETER SENSITIVITY

In this section, we explore the effect of hyperparameters of our proposed FedeRiCo.

**Effect of $\epsilon$-greedy sampling** Here we show the effect of different $\epsilon$ values. Recall that each client deploys an $\epsilon$-greedy selection strategy. The smaller the value of $\epsilon$, the more greedy the client is in selecting the most relevant collaborators with high weights, leading to less exploration. Fig. 6a shows the accuracy along with training rounds with different $\epsilon$ values on the Office-Home dataset. One can see that there is a trade-off between exploration and exploitation. If $\epsilon$ is too high (e.g., $\epsilon = 1$, uniform sampling), then estimates of the likelihoods/losses are more accurate. However, some gradient updates will vanish because the client weight is close to zero (see Section 3.3), resulting

---

[4]We used uniform sampling for Fig. 5a ($\epsilon = 1$) as most of the client weights are 0 after a few rounds.

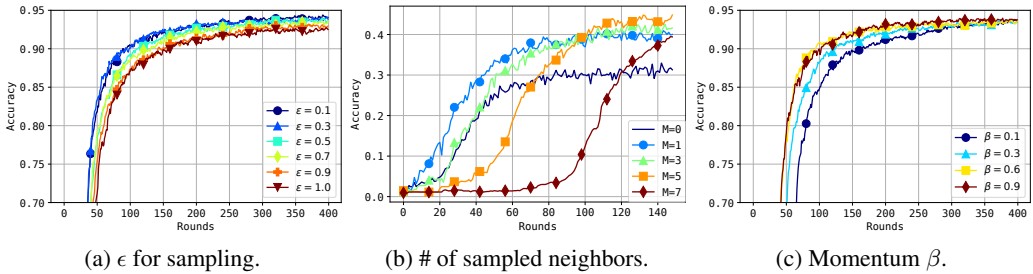

(a) $\epsilon$ for sampling.     (b) # of sampled neighbors.     (c) Momentum $\beta$.

Figure 6: Test accuracy with different hyperparameters.

in slow convergence. On the other hand, if $\epsilon$ is too small, the client may miss some important collaborators due to a lack of exploration. As a result, we use a moderate $\epsilon = 0.3$ in all experiments.

**Effect of number of sampled neighbors** We plot accuracy with number of neighbors $M \in \{0, 1, 3, 5, 7\}$ on CIFAR100 with 4 different client distributions, where $M = 0$ is similar to Local Training as no collaboration happens. As shown in Fig. 6b, when the number of neighbors increases, FedeRiCo converges more slowly as each client is receiving more updates on other client's models. While a smaller number of neighbors seems to have a lower final accuracy, we notice that even with $M = 1$, we still observe significant improvement compared to no collaboration. Therefore, we use $M = 3$ neighbors in our experiments as it has reasonable performance and communication cost.

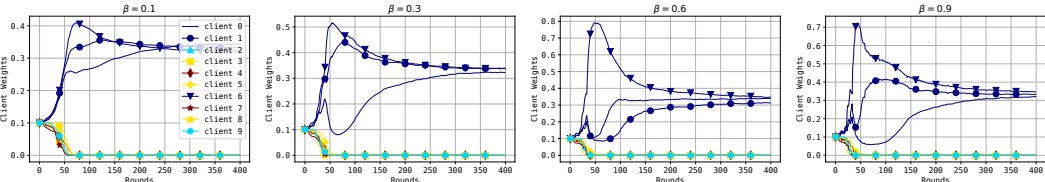

Figure 7: Client weights (client 0) with different momentum values $\beta$ on the client weight update.

**Effect of client weight momentum** We plot the overall test accuracy of client 0 on the Office-Home dataset with 4 different data distributions over $\beta \in \{0.1, 0.3, 0.6, 0.9\}$ in Fig. 6c and similarly for the client weights in Fig. 7. With smaller $\beta$, as shown in Fig. 7, we observe a smoother update on the client weights, which is expected as the old tracking loss dominates the new one. Although various values produce similar final client weights, a bigger $\beta$ can lead to more drastic changes in early training. However, one shouldn't pick a very small $\beta$ just because it can produce smoother weights. As shown in Fig. 6c, the algorithm may converge more slowly with smaller $\beta$. Therefore, we use $\beta = 0.6$ as it encourages smoother updates and also maintains good convergence speed.

## 5   CONCLUSION AND DISCUSSION

In this paper, we proposed FedeRiCo, a novel framework for decentralized and personalized FL derived from EM for non-i.i.d client data. We evaluated FedeRiCo across different datasets and demonstrated that FedeRiCo outperforms multiple existing personalized FL baselines and encourages clients to collaborate with similar clients, i.e., the right collaborators.

Decentralized FL provides an alternative architecture in the absence of a commonly trusted server. For example, CB-DEM ()forero2008consensus studies the distributed EM algorithm for classification in wireless networks with local bridge sensors to ensure all sensors reach consensus. Compared to our decentralized communication-efficient protocol, CB-DEM requires global information every round for the consensus synchronization. While decentralized FL removes the risk of single point failure compared to centralized FL by using peer-to-peer communication, it also raises concerns about security risks with the absence of a mutually trusted central server. Therefore, a promising direction is to incorporate trust mechanisms long with decentralization (Kairouz et al., 2019), such as blockchain frameworks (Qin et al., 2022). Additionally, FL schemes do not provide an explicit guarantee that private information will not be leaked. However, most FL frameworks, including ours, are compatible with privacy-preserving training techniques such as differential privacy, which is another promising and interesting research direction (Wei et al., 2020; Truex et al., 2020).

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
