# OpenReview forum: "Find Your Friends: Personalized Federated Learning with the Right Collaborators"
_ICLR.cc/2023/Conference — Submitted to ICLR 2023_

### Official Review · Reviewer_r1RL · 2022-10-24

**Confidence:** 4
**Correctness:** 3
**Technical Novelty And Significance:** 2
**Empirical Novelty And Significance:** 3
**Recommendation:** 6

**Clarity, Quality, Novelty And Reproducibility:**

This paper proposed a technically sound approach to tackle data heterogeneity. It is clearly written with moderate novelty.

**Strength And Weaknesses:**


Pros:
+ Technical sound approach towards addressing data heterogeneity.
+ Well-designed experiments that validate each component in the proposed approach.
+ Comprehensive related work.

Cons:

Lowe efficiency in 1) communication during learning and 2) model inference after learning convergence. Even with the modifications that relax the all-to-all communication pre-requisite, each client still needs to download gradients from multiple neighbors. While FedAvg style algorithm is more communication efficient. The final model inference still requires a forward pass through multiple models. In general, I feel that this design is not practical when the model size is scaled up or when the number of users is high.

Minor:

An algorithm box should be provided to summarize the proposed approach.


**Summary Of The Paper:**

This paper tackles the data heterogeneity issue in Federated Learning (FL) by letting local clients leverage model parameters from nodes with similar data distributions. Each client approximates the data similarity of its neighbors through a weighing parameter $w_{ij}$ which is learned by EM optimization. In order to achieve this learning, each client needs to broadcast its learned model gradients to (selected) neighbors. The weighted ensemble of neighboring models is used for inference as the final model once upon training convergence.


**Summary Of The Review:**

This paper proposed an FL approach that ensemble local models with similar data distributions to tackle data heterogeneity. This approach is technically sound but could raise potential concerns regarding communication efficiency and scalability.

---

> ### Author Response · Authors · 2022-11-15
> **Response to Reviewer r1RL**
>
> Thank you for your time spent reviewing our work. We are glad you found our work to be technically sound, with well-designed experiments and a comprehensive review of related work. We are happy to address your remaining concerns below.
>
> **1.Low efficiency in communication during learning. Even with the modifications that relax the all-to-all communication prerequisite, each client still needs to download gradients from multiple neighbors.**
>
> We appreciate your relevant concern about communication efficiency. We agree that communication cost could be a bottleneck in not only our algorithm but decentralized Federated Learning in general. By relaxing the all-to-all communication, the scalability of our framework is enhanced. We have shown in Fig.6b that our method can work well even when each client only communicates with one neighbor, which incurs the same communication cost as centralized settings like FedAvg.
>
> **2.Low efficiency in model inference after learning convergence. The final model inference still requires a forward pass through multiple models. In general, I feel that this design is not practical when the model size is scaled up or when the number of users is high.**
>
> We thank the reviewer for the valuable feedback. We agree that when there are multiple relevant clients (good collaborators), inference can be computationally heavy as it would involve forward passing through multiple models. However, this can be addressed by either of the following two possible solutions:
>
> 1. For every client, sample k models (based on client weights) for prediction. Theoretically, we would get an unbiased estimate of the prediction but with higher variance. In the extreme case, one may even use only the local model for prediction, or generally a single model chosen greedily based on the learned client weights.
>
> 2. As introduced in section 3.3 under the heading ``Tracking the losses for the posterior”, we proposed an exponential moving average to encourage client collaboration. In section 4.4, we show that the cumulative loss results in sparse, one-hot like client weights whereas the exponential moving average loss results in less sparse client weights. Therefore, to reduce the computation overhead during inference, we could adopt the exponential moving average loss in the early stage of training to encourage new client collaboration and use cumulative loss in the later stage to encourage sparser final client weights. With sparse client weights, inference will depend on a much smaller subset of client models.
>
> **3.An algorithm box should be provided to summarize the proposed approach.**
>
> We also thought that an algorithm box would help readers to understand our approach. Due to space constraints, we had originally included the algorithm box in Appendix B in the Supplementary Material.
>
> We hope that our responses have addressed your concerns. If you have any more questions, we will be happy to answer them.

---

### Official Review · Reviewer_Knpq · 2022-10-26

**Confidence:** 5
**Clarity, Quality, Novelty And Reproducibility:** The paper is written clearly and nove…
**Correctness:** 4
**Technical Novelty And Significance:** 3
**Empirical Novelty And Significance:** Not applicable
**Recommendation:** 6

**Strength And Weaknesses:**

*Strengths*
- The paper is well written and easy to follow.
- The problem is interesting and proposed solution is discussed theoretically as well as experimentally.

*Weakness*

- I am not sure if decentralized setting is well motivated for federated learning. This is specially be because the FL is specifically for larger number of devices, and implementing a decentralized algorithm is almost impossible in those settings. The authors should provide some specific use cases for the motivation.

- The proposed framework seems to solve the issue of FedAvg, but it brings is new issue of requirement of agents communicating with each other which can be impractical to consider. Please provide some examples.

- How to select the neighbors, what kind of communication graph is required for the algorithm to work?

- What are the additional challenges in the theoretical analysis for the decentralized settings? For instance, what are the additional challenges need to be addressed as compared to decentralized EM algorithm in the following paper, and it's related papers.

P. A. Forero, A. Cano and G. B. Giannakis, "Consensus-based distributed expectation-maximization algorithm for density estimation and classification using wireless sensor networks," 2008 IEEE International Conference on Acoustics, Speech and Signal Processing, 2008, pp. 1989-1992, doi: 10.1109/ICASSP.2008.4518028.


**Summary Of The Paper:**

This paper focuses on personalized server free federated learning. A decentralized algorithm is proposed based on EM algorithm in which each device learns from the devices which has similar data distributions. Theoretical results are provided along with empirical justifications.

**Summary Of The Review:**

Overall the paper is written well, and the idea is interesting, but it is not clear how much is the contribution in terms of algorithmic development as compared to existing decentralized EM algorithms in the literature.

---

> ### Author Response · Authors · 2022-11-15
> **Response to Reviewer Knpq**
>
> Thank you for your positive review! We are glad you found the problem we tackled to be interesting and our solution novel. Hopefully we can address your remaining concerns below.
>
> **1.I am not sure if decentralized setting is well motivated for federated learning. This is specially because the FL is specifically for larger number of devices, and implementing a decentralized algorithm is almost impossible in those settings. The authors should provide some specific use cases for the motivation.**
>
> We appreciate the valuable comment for use cases, but we disagree that FL is specifically for a large number of devices, and we don’t believe decentralized algorithms are impossible in such settings. We would also like to note that reviewers **xqTG** and **Tffe** both highlighted that our problem setting was well-motivated.
>
> One specific use case for decentralized FL is the cross-silo FL setting (Qiang et al. 2019) where big institutions (banks, hospitals, etc.) may wish to collaborate but can not directly share data out of privacy and regulatory concerns. Additionally, it may be challenging to find a trustworthy third party as the central server due to heavy regulations or the desire for autonomy, rendering centralized FL infeasible. Big institutions have reliable connections, making it feasible to implement a decentralized algorithm.
>
> With regards to decentralized algorithms for a larger number of clients, there is significant interest in this space. In practice, a commonly trusted central server is not always easily accessible and decentralization techniques are required. A promising and popular direction for decentralized FL is to adopt blockchain mechanisms (Nguyen et al. 2021, Lu et al. 2020). The success of decentralized blockchains in FL clearly shows that decentralized algorithms are not impossible in this setting.
>
> Yang, Qiang, et al. "Federated learning." Synthesis Lectures on Artificial Intelligence and Machine Learning 13.3 (2019): 1-207
> D. C. Nguyen et al., "Federated Learning Meets Blockchain in Edge Computing: Opportunities and Challenges," in IEEE Internet of Things Journal, vol. 8, no. 16, pp. 12806-12825, 15 Aug.15, 2021, doi: 10.1109/JIOT.2021.3072611.
>
> Y. Lu, X. Huang, Y. Dai, S. Maharjan and Y. Zhang, "Blockchain and Federated Learning for Privacy-Preserved Data Sharing in Industrial IoT," in IEEE Transactions on Industrial Informatics, vol. 16, no. 6, pp. 4177-4186, June 2020, doi: 10.1109/TII.2019.2942190.
>
> **2.The proposed framework seems to solve the issue of FedAvg, but it brings a new issue of requirement of agents communicating with each other which can be impractical to consider. Please provide some examples.**
>
> An example use case is the cross-silo FL we mentioned above, where all clients have strong computing power, and can communicate the model parameters to each other instead of to the central server. Such a central server may not be available in heavily regulated domains such as for financial and medical institutions: one hospital may exchange limited information with another hospital, but may not communicate with an unauthorized third party. We believe that enabling clients to communicate directly with each other can bring significant benefits. Besides reducing single-point failure, peer-to-peer connections can reduce the processing bottleneck incurred at the central server, which usually needs to process requests from *all* clients simultaneously, and can reduce the required communication bandwidth.
>
> We want to emphasize that the main focus of this paper is building **personalized** models. Communication, whether with the central server or with other clients, is unavoidable in collaborative learning. One can easily adapt our method to the centralized FL setting, but this merely shifts the brunt of the communication costs to a different party, it does not eliminate them.
>
> **3.How to select the neighbors, what kind of communication graph is required for the algorithm to work?**
>
> In our experiments, without making any assumption about the client connections, we adopt $\epsilon$-greedy sampling for choosing the neighbors. When $\epsilon$ is close to 1, sampling becomes closer to uniform, and when $\epsilon$ is close to 0, sampling is greedy (according to the client weights). See Section 3.3 under the heading ``Sampling Scheme”.
>
> Our framework is not restricted to any particular communication graph. As long as information can flow through all clients in a finite number of FL rounds, our proposed algorithm will work in theory. One possible way to specify the communication graph is via the PushSum scheme (Mahmoud, et al. 2019) which would be an interesting direction for future research.
>
> Assran, Mahmoud, et al. "Stochastic gradient push for distributed deep learning." International Conference on Machine Learning. PMLR, 2019.

---

### Official Review · Reviewer_Tffe · 2022-10-26

**Confidence:** 2
**Clarity, Quality, Novelty And Reproducibility:** Clarity is good. The algorithm is nov…
**Correctness:** 4
**Technical Novelty And Significance:** 4
**Empirical Novelty And Significance:** 3
**Recommendation:** 6

**Strength And Weaknesses:**

**Strength**
1. The problem is well motivated.
2. This paper proposed a novel algorithm (to my best knowledge, though I am not very familiar with the literature) with the guarantee of convergence.

**Weaknesses**
1. According the description of dataset, is it possible that when clients are in the sample group, some data will simultaneously appear in the training data (of some client) and test data (of other client)?
2. The paper seems to miss the experiment that different distributions have some overlap instead of disjoint support, e.g. client 1 with 80% label 0 and 20% label 1 and client 2 with 20% label 0 and 80% label 1? When this happens, for sanity check, would the mixtures of distributions by learned weights are the distributions of each client?

**Summary Of The Paper:**

This paper studies the problem that the global model may perform poorly for most clients, e.g. the data at different clients is very diverse. It addresses this problem by learning how data distributions of different clients mix together as the distribution at client i. It utilizes EM algorithm to keep updating the mixture weights and in the end each client makes predictions by a mixture of all clients' local models. Further, it proposes a version with efficient computation. In the experiments, the efficacy of proposed algorithms is justified.

**Summary Of The Review:**

The problem and the proposed algorithm are well motivated. However, experiments at one reasonable setting are missed.

---

> ### Author Response · Authors · 2022-11-15
> **Response to Reviewer Tffe**
>
> Thank you for your time spent reviewing, we are glad you found our work novel and well-motivated. We are happy to address your concerns below.
>
> **1.According to the description of dataset, is it possible that when clients are in the sample group, some data will simultaneously appear in the training data (of some client) and test data (of other client)?**
>
> No, in our experiments clients received disjoint local datasets and then further split their datasets locally into train and test sets. The same data point never appeared twice in our setting.
>
> **2.The paper seems to miss the experiment that different distributions have some overlap instead of disjoint support, e.g. client 1 with 80% label 0 and 20% label 1 and client 2 with 20% label 0 and 80% label 1? When this happens, for sanity check, would the mixtures of distributions by learned weights are the distributions of each client?**
>
> This is a good question. We explicitly used a disjoint data split in our experiments in order to have a ``ground truth” of optimal collaborations among clients so that we could check if our method was identifying such collaborations. We agree that overlapping client classes is a valid setting, which is why we had originally included an experiment in Appendix C with the Dirichlet split from FedEM (Marfoq et al. 2021) where clients have overlapping classes. As shown in Table 2, FedeRiCo still outperforms all baseline methods.
>
> For the sanity check you proposed, we would like to emphasize our theoretical assumption where each client dataset $D_i$ can be viewed as a mixture of $K$ distributions, where $K$ is the number of clients. In other words, the base distributions in the mixture are client distributions rather than class distributions. In the proposed setting where client 1 has 80%/20% label 0/1 and client 2 has 20%/80% label 0/1, what matters is that their underlying distributions are not the same. It is not expected that our method would result in client 1 learning weights of 0.8 on itself, and 0.2 on client 2 for example.
>
> We hope that our responses have addressed your concerns. If you have any more questions, we will be happy to answer them.

---

### Official Review · Reviewer_xqTG · 2022-10-29

**Confidence:** 3
**Correctness:** 3
**Technical Novelty And Significance:** 2
**Empirical Novelty And Significance:** 2
**Recommendation:** 3

**Clarity, Quality, Novelty And Reproducibility:**

- Clarity: the presentation of the paper is clear, and the proposed idea is easy to follow.
- Quality: the writing quality of the paper is good. Both theoretical and experimental results are shown to justify the effectiveness of the proposed method.
- Novelty: the proposed method seems to be somewhat known, and thus the novelty is limited.
- Reproducibility: enough details are shown for the implementation and experimental setup. I thus believe the experiments can be reproduced.

**Strength And Weaknesses:**

Strength:
- the paper is well-written and the targetted problem is well motivated.
- both theoretical and empirical results are shown to demonstrate the effectiveness of the proposed method.

Weaknesses:
- the proposed EM algorithm seems to be somewhat known, which limits the novelty of the proposed FedeRiCo method.
- the privacy guarantees for the decentralized federated learning framework are not clear. It seems the individual client's gradient can be seen by the other clients (in the small gradient steps in lieu of full optimization in each round). The authors are expected to discuss the privacy guarantees more explicitly.
- the scale of the experiment is small, how does FedeRiCo perform on the ImageNet scale datasets?

**Summary Of The Paper:**

The paper proposes FedeRiCo, a decentralized federated learning framework that allows clients to collaborate much or little with other participating clients to enhance the performance of the federated learned models. FedeRiCo leverages an EM-style optimization procedure to solve the global model weights and clients' collaboration weights concurrently. Both theoretical analysis and empirical results are shown to justify the effectiveness of the proposed FedeRiCo method.

**Summary Of The Review:**

The paper seeks to tackle an interesting and important problem in federated learning to enhance collaboration among participating clients. The proposed method makes sense but is somewhat known. I am thus concerned about the novelty of the paper. Moreover, the privacy guarantees of the proposed method are not clear, which is a clear weakness.

---

> ### Author Response · Authors · 2022-11-15
> **Response to Reviewer xqTG**
>
> Thank you for your helpful comments. We are glad that you found our paper well-written, well-motivated, and that our theoretical and empirical results demonstrate the effectiveness of our proposal. We will address your concerns point-by-point.
>
> **1. The proposed EM algorithm seems to be somewhat known, which limits the novelty of the proposed FedeRiCo method.**
>
> We appreciate your concern, however, reviewers Tffe and Knpq highlighted our proposed algorithm as having good novelty compared to the existing literature. May we kindly ask you to be more specific in how it is ``somewhat known” so we can address your concern more precisely? Are there other relevant references that we did not discuss?
>
> The closest work based on EM is FedEM (Marfoq et al, 2021), with which we extensively compared and contrasted our proposed method. We explained how FedEM uses instance-level weight assignment in training, but a client-level weight assignment for inference, a mismatch which prevents FedEM from correctly identifying the distribution a new data point was generated from. In this regard, our method corrects the mismatch and uses assumptions on the data distribution that are more realistic, which translates to the better empirical performance shown in our experiments.
>
> **2.The privacy guarantees for the decentralized federated learning framework are not clear. It seems the individual client's gradient can be seen by the other clients (in the small gradient steps in lieu of full optimization in each round). The authors are expected to discuss the privacy guarantees more explicitly.**
>
> Privacy is an important consideration, and we agree that privacy is a key motivation of FL. Recall that centralized FL protects privacy by exchanging model parameters instead of raw data, and hence clients must still trust the central server with their models. In the same way, decentralized FL also only shares model parameters, but clients need to trust other clients. Furthermore, model parameters can still leak private information about the raw data, hence, FL alone does not provide an explicit/theoretical privacy guarantee. However, most FL frameworks, including ours, are compatible with privacy-preserving training techniques such as differential privacy. We see this as another promising research direction, but it is orthogonal to our main focus of personalization.
>
> Because privacy is an important motivation for introducing FL systems, we have included a paragraph discussing this in the revised version, Section 5.
>
> **3.The scale of the experiment is small, how does FedeRiCo perform on the ImageNet scale datasets?**
>
> We followed the convention of most existing Federated Learning papers and use standard CIFAR datasets (Marfoq et al. 2021, Sattler et al. 2020, Zhang et al. 2021, McMahan et al. 2017, Shen et al. 2020, Huang et al. 2021, Ghosh et al. 2020, Arivazhagan et al. 2019, Acar et al. 2021). We believe that our proposed algorithm would also work for larger datasets, however, testing this requires significantly more computational resources to compare against all other baseline methods since many, if not all of them, did not benchmark on the ImageNet dataset.
>
> We appreciate that you evaluated our paper to be good in three out of the four criteria (Clarity, Quality, and Reproducibility), but we found the recommendation of ``Reject” to be inconsistent with this praise. We kindly ask you to elaborate more on your concerns about novelty, and consider adjusting your score if our response helps to address your concerns.

---

### Author Response · Authors · 2022-11-15
**General Response**


We thank all four reviewers for their insightful feedback and constructive reviews. We were pleased to see that reviewers appreciated the combination of theoretical and experimental results (**xqTG**, **Knpq**), and that they thought our work was novel (**Tffe**, **Knpq**, **r1RL**), well-written and clear (**xqTG**, **Tffe**, **Knpq**, **r1RL**), and that the problem was well-motivated and interesting (**xqTG**, **Tffe**, **Knpq**). We will respond to each review separately.

---

### Author Response · Authors · 2022-11-17
**Reminder for discussion**

We kindly remind our reviewers that the discussion period comes to an end in one day. Do you have any further concerns we can address?

---

### Decision · Program_Chairs · 2023-01-20

**Decision:**

Reject

**Justification For Why Not Higher Score:**

See above. The problem setting is questionable and almost all reviewers brought up the issue. AC echos.

**Justification For Why Not Lower Score:**

N/A

**Metareview: Summary, Strengths And Weaknesses:**

This paper focuses on a new server-free decentralized setting of federated learning. Their algorithm is based on an EM algorithm in which each device learns from other devices with similar data distributions. All reviewers appreciate the clarity of the paper writing and the theoretical support for the algorithm.

However, several reviewers expressed concern regarding the decentralized FL setting. As the authors also agreed with, their new setting (1) does not scale up conveniently w.r.t. massive clients, due to explosive cross-client communication overhead in this way; and (2) might need all clients to have strong computing power. The authors argued the validity of this setting by citing examples such as a handful of big institutions (banks, hospitals, etc.) collaborating with each other. This is maybe plausible but is completely not reflected by the current experiments that use nothing but all centralized FL's classical benchmarks (CIFAR etc). Hence, there is at least some disconnection between the authors' promised motivation story and their delivered experiments.

Also, the AC also thinks the novelty of proposing EM algorithm in FL setting was somehow exaggerated. For example, there have been several prior arts combining FL and mixture-of-experts (MoEs), between 2020 and 2022. They are certainly not the same as the proposed algorithm, but MoEs can be viewed as an EM variant too - and they often put client-level weight assignment as well. The authors are invited to take broader literature into positioning their work.

Overall, the paper is promising but cannot be accepted in its current shape. The AC encourages authors to think more carefully of their work positioning (in both aforementioned ways) and wishes authors the best of luck in their next try.